# Cobalt Phthalocyanine-Doped Polymer-Based Electrocatalyst for Rechargeable Zinc-Air Batteries

**DOI:** 10.3390/ma16145105

**Published:** 2023-07-20

**Authors:** Yogesh Kumar, Srinu Akula, Elo Kibena-Põldsepp, Maike Käärik, Jekaterina Kozlova, Arvo Kikas, Jaan Aruväli, Vambola Kisand, Jaan Leis, Aile Tamm, Kaido Tammeveski

**Affiliations:** 1Institute of Chemistry, University of Tartu, 50411 Tartu, Estonia; yogesh.kumar@ut.ee (Y.K.); srinu.akula@ut.ee (S.A.); elo.kibena-poldsepp@ut.ee (E.K.-P.); maike.kaarik@ut.ee (M.K.); jaan.leis@ut.ee (J.L.); 2Institute of Physics, University of Tartu, 50411 Tartu, Estonia; jekaterina.kozlova@ut.ee (J.K.); arvo.kikas@ut.ee (A.K.); vambola.kisand@ut.ee (V.K.); aile.tamm@ut.ee (A.T.); 3Institute of Ecology and Earth Science, University of Tartu, 50409 Tartu, Estonia; jaan.aruvali@ut.ee

**Keywords:** phloroglucinol-formaldehyde network, nitrogen doping, electrocatalysis, non-precious metal catalyst, oxygen reduction reaction, oxygen evolution reaction, rechargeable zinc-air battery

## Abstract

Rechargeable zinc-air batteries (RZAB) have gained significant attention as potential energy storage devices due to their high energy density, cost-effectiveness, and to the fact that they are environmentally safe. However, the practical implementation of RZABs has been impeded by challenges such as sluggish oxygen reduction reaction (ORR) and oxygen evolution reaction (OER), including poor cyclability. Herein, we report the preparation of cobalt- and nitrogen-doped porous carbon derived from phloroglucinol-formaldehyde polymer networks with 2-methyl imidazole and cobalt phthalocyanine as precursors for nitrogen and cobalt. The CoN-PC-2 catalyst prepared in this study exhibits commendable electrocatalytic activity for both ORR and OER, evidenced by a half-wave potential of 0.81 V and *E*_j=10_ of 1.70 V. Moreover, the catalyst demonstrates outstanding performance in zinc-air batteries, achieving a peak power density of 158 mW cm^−2^ and displaying excellent stability during charge-discharge cycles. The findings from this study aim to provide valuable insights and guidelines for further research and the development of hierarchical micro-mesoporous carbon materials from polymer networks, facilitating their potential commercialisation and widespread deployment in energy storage applications.

## 1. Introduction

Rechargeable zinc-air batteries (RZABs) have emerged as a promising technology for high-energy density-based energy storage systems. However, their practical implementation has been hindered by challenges such as sluggish oxygen reduction and evolution reactions, as well as limited cyclability [1,2]. To overcome these limitations, extensive research efforts have been focused on developing efficient catalysts to enhance the electrochemical performance of the RZABs [3,4,5,6]. Presently, precious metal-based materials (such as Pt, Pd, Ru, and Ir) are used as electrocatalysts, which has hindered their use in practical applications due to their high price, limited resources, and limited stability [7,8,9]. Transition metal-derived catalysts have gained significant attention due to their unique properties, including their abundance, high electrocatalytic activity, tuneable composition, and excellent stability compared to noble metal-based catalysts [10,11,12,13].

Carbon support also plays a vital role in enhancing catalysts’ overall electrocatalytic performance by integrating the active centres’ distribution and promoting the efficient mass transport of reactant species. There are several methods for preparing porous carbon supports, including pyrolysis of metal-organic frameworks (MOFs), covalent organic frameworks (COFs), polymers, bio-mass, and so on [6,14,15,16,17]. However, porous carbon materials synthesised using these methods suffer severe degradation at high voltage conditions in RZABs due to their insufficient graphitisation levels [18,19,20]. Thus, porous carbon-based catalysts derived from the rigid polymer precursors embedded transition metal dopants that undergo controlled pyrolysis result in desirable structures, compositions, and electrocatalytic properties [21,22,23]. Moreover, porous carbon derived from polymer-network materials possesses tuneable textural properties, high surface area, and good electrical conductivity [24]. Transition metal insertion into nitrogen-rich polymer networks offers more opportunities for enhancing catalytic activity and stability, thus improving the overall performance of RZABs [25,26]. For instance, Zhu et al. developed a cobalt-doped nitrogen-containing carbon material derived from a polydopamine precursor. They demonstrated excellent electrocatalytic activity for ORR in zinc-air batteries [27]. The Co-doped carbon material exhibited enhanced electrochemical performance with high stability and remarkable resistance to carbon corrosion by mitigating the peroxide formation. Chen et al. synthesised an MnO_x_-anchored carbon nanotubes embedded FeCo alloy that showed an efficient OER in RZABs due to its extraordinary selectivity [28]. In addition to cobalt and manganese, other transition metals such as iron, nickel, and copper have also been investigated as dopants to polymer-derived catalysts for oxygen electrocatalysis [23,29,30,31].

The transition metal can also be introduced in the carbon matrix using metal macrocycles (e.g., metal porphyrins, phthalocyanines) [32,33,34,35]. Metal phthalocyanines have a unique mix of electrical properties and versatile structures that may be used to create metal-coordinated nitrogen (M-Nx) sites, which are highly favourable for ORR [36]. Despite the progress made in utilising transition metal-doped polymer-derived catalysts for RZABs, there are still challenges to be addressed to further improve their performance. Optimisation of the doping concentration, pyrolysis conditions, and precursor composition is crucial for achieving desired electrocatalytic properties via precisely controlling the textural properties. Moreover, understanding the catalytic mechanisms and the impact of transition metal doping on the structure–activity relationships of polymer-derived catalysts is essential for further enhancements.

Herein, we prepared Co and N-doped porous carbon materials derived from the polycondensation of phloroglucinol with formaldehyde via the formation of a polymer framework; cobalt phthalocyanine and 2-methylimidazole are used as metal and nitrogen sources. These materials were prepared at different temperatures (700, 800, 900 °C) and metal contents to assess their optimised ORR/OER electrocatalytic activity in 0.1 M KOH using the rotating disk electrode (RDE) method. CoN-PC-2 prepared at 800 °C exhibits the best ORR/OER activity among the prepared catalysts, with *E*_1/2_ and *E*_j=10_ of 0.81 V and 1.70 V, respectively. The morphology, composition, and structure of the catalyst materials were characterised by scanning electron microscopy (SEM), X-ray diffraction (XRD), X-ray photoelectron spectroscopy (XPS), and N_2_ physisorption analysis. Finally, catalysts were tested in an RZAB configuration as cathode catalysts, and the CoN-PC-2 catalyst delivered the highest peak power density with good stability due to its hierarchical texture and optimised metal content.

## 2. Materials and Methods

### 2.1. Materials

Phloroglucinol (99%, Alfa Aesar, Haverhill, MA, USA), formaldehyde solution (37%, Thermo Fisher Chemical, Waltham, MA, USA), and cobalt phthalocyanine (CoPc, 99%, Alfa Aesar, Haverhill, MA, USA) were used as received. Five wt% Nafion solution and potassium hydroxide (85%) were obtained from Sigma-Aldrich. Commercial Pt/C (20%, E-TEK) and RuO_2_ (99%, Sigma Aldrich, St. Louis, MO, USA) were used for comparison. Glassy carbon (GC, GC-20SS, Tokai Carbon, Tokyo, Japan) was used as electrode substrate for electrochemical measurements. In this study, Milli-Q water (18 MΩ cm) was employed for the preparation of solutions.

### 2.2. Synthesis of Phloroglucinol-Formaldehyde-Methylimidazole (PFM) Framework

For a standard synthesis procedure, 2.2 g of phloroglucinol was dissolved in 20 mL of Milli-Q water in a beaker at a temperature of 45 °C. Then, 10 g of formaldehyde solution was mixed in phloroglucinol solution under continuous stirring for 30 min. The reaction between phloroglucinol and formaldehyde took place, forming a yellowish-orange precipitate. Next, 4.4 g of 2-methylimidazole was dissolved in another beaker containing 40 mL water at 45 °C. Then, both solutions were mixed and stirred for 12 h at 60 °C. Subsequently, the product underwent filtration, followed by multiple water rinses and drying at a temperature of 60 °C, resulting in the production of the PFM framework.

### 2.3. Synthesis of CoN-PC

For the synthesis of CoN-PC-1, 100 mg of PFM powder and 5 mg of CoPc were suspended in 25 mL of ethanol by sonication for 1 h. Then, the mixture was dried out at 60 °C followed by pyrolysis at 800 °C under an inert atmosphere to get the CoN-PC-1. The CoN-PC-2 was also prepared via a similar route by changing the amount of CoPc to 10 mg and PC-800 was prepared by pyrolysing the PFM polymer at 800 °C without adding CoPc.

### 2.4. Physical Characterisation

The surface morphology of the prepared catalysts was assessed using a Helios Nanolab 600, FEI (Hillsboro, OR, USA) scanning electron microscope (SEM). The elemental mapping of the prepared catalysts and the determination of bulk elemental composition were performed by the energy dispersive X-ray spectrometer (EDX) from Oxford Instruments (Abingdon, UK), coupled to the scanning electron microscope. Catalyst suspension was drop-casted onto a polished glassy carbon (GC) disk to prepare SEM samples. The catalysts’ crystallinity was determined through X-ray diffraction (XRD) analysis. The XRD measurements were conducted on a Bruker D8 Advance diffractometer (Billerica, MA, USA) with Ni-filtered Cu Kα (λ = 1.54184 Å) radiation. Scanning steps of 0.0126° were used to obtain the diffraction patterns over a 2θ range of 5–90°.

For N_2_ physisorption experiments, the catalyst materials were dried in vacuum overnight at 300 °C prior to analysis. The NovaTouch LX2 instrument (Quantachrome) was used for the analysis, and TouchWin 1.11 software was used for all calculations. The specific surface area (S_BET_) was calculated utilising the Brunauer–Emmett–Teller method in the *P*/*P*_0_ range of 0.02–0.2. The quenched solid density functional theory (QSDFT) equilibria model for slit-type pores was utilised for the calculation of the volume of micropores (V_µ_), and total pore volume (V_tot_) was evaluated at the N_2_ saturation pressure (*P*/*P*_0_ = 0.97).

X-ray photoelectron spectroscopy (XPS) was used to determine the surface elemental composition using the electron energy analyser SCIENTA SES 100 and Mg K*α* X-rays (1.2536 keV) from Thermo XR3E2, a twin-anode X-ray tube. The survey scan was measured in the range of 1000 to 0 eV with a 0.5 eV step size, 0.2 s step duration, and an average of five scans. The core-level spectra were averaged with a 0.2 eV step size and 0.2 s step duration over the course of thirty scans. The Gauss–Lorentz hybrid function (GL 70, Gauss 30%, Lorentz 70%) was utilised for the data analysis on CasaXPS software (version 2.3.17).

### 2.5. Electrochemical Characterisation

The electrochemical ORR and OER measurements were performed at room temperature with a potentiostat/galvanostat PGSTAT30 (Metrohm-Autolab, Utrecht, The Netherlands) in a conventional three-electrode system using a platinum wire, saturated calomel electrode (SCE), and catalyst-coated GC rotating disk electrode (geometric area = 0.196 cm^2^) as a counter electrode, reference electrode, and the working electrode, respectively. A polished GC disk was drop-cast with 10 µL of a homogeneous suspension of 4 mg of prepared catalyst in 1000 µL of 2-propanol: water: Nafion (1: 1: 0.03) and dried at 60 °C to achieve the 0.2 mg cm^−2^ catalyst loading.

All potentials were converted against the reversible hydrogen electrode (RHE) using the equation *E*_RHE_ = *E*_SCE_ + 0.242 + 0.059 × pH, and the electrochemical data given in this work were *i*R-corrected using the positive feedback method. To measure the background current, cyclic voltammograms (CV) were recorded in an Ar-saturated 0.1 M KOH solution at a potential scan rate (*v*) of 10 mV s^−1^. Furthermore, the rotating disk electrode (RDE) technique was used to investigate the ORR and OER activity of the catalysts. To change the electrode rotation rate (between 360 and 3100 rpm), a speed control unit (CTV101) was attached to a rotator (EDI101, Radiometer). Measurements of the OER polarisation curves were carried out in an Ar-saturated 0.1 M KOH at a 1900 rpm rotation rate and a scan rate of 10 mV s^−1^. *E*_j=10_ is denoted as the potential at which the OER current density reaches 10 mA cm^−2^. For ORR stability, the working electrode was cycled 10,000 times between 1.0 and 0.6 V vs. RHE at 100 mV s^−1^ in O_2_-saturated 0.1 M KOH solution. Before and during the stability test, the RDE polarisation curves were measured at 1900 rpm at 10 mV s^−1^. For the OER stability test, the working electrode was held at 1.7 V at 1900 rpm for 3 h.

### 2.6. RZAB Test

The RZAB performance and stability evaluations of the electrocatalysts were assessed using a homemade battery device. A 6 M KOH solution containing 0.2 M Zn(CH_3_COO)_2_ was used as the electrolyte for the RZABs. Seven mg of catalyst was dispersed by sonicating in 800 µL water-ethanol (1:3) solution along with 20 µL Nafion solution as a binder to produce a homogenous catalyst ink. The catalyst ink was drop-casted onto the 3.5 cm^2^ carbon paper (Sigracet 39 BB) to obtain the catalyst loading of 2 mg cm^−2^. The anode was made of a polished zinc sheet with a 0.2 mm thickness. The exposed surface area to the electrolyte solution for both the electrodes-Zn sheet and the catalyst-coated carbon paper was 0.79 cm^2^. The assembled RZABs’ stability was assessed using a 10 min galvanostatic discharge-charge cycle at 5 mA cm^−2^.

## 3. Results and Discussion

### 3.1. Physico-Chemical Characterization

XRD analysis was used to investigate the catalyst materials’ crystalline structure (Figure 1a). All XRD patterns depicted two broad peaks around 26° and 43.5°, corresponding to the (002) and (100) graphitic planes, respectively. No diffraction peaks were detected corresponding to metals, indicating the absence of large Co nanoparticles in the prepared catalysts. PC-800 showed slightly higher intensity peaks than the Co-doped materials, pointing out that the introduction of transition metal tends to modify the graphitic structure [37]. Further, the N_2_ physisorption analysis was carried out to study the textural properties of the materials (Figure 1b and Table 1). Typical type-II adsorption-desorption isotherms with H3 hysteresis showed the presence of micropores and mesopores in all prepared catalysts. The specific surface area (S_BET_) of PC-800, CoN-PC-1, and CoN-PC-2 was 449, 464, and 325 m^2^ g^−1^. PC-800 displayed the highest total porosity and microporosity, and both tend to decrease with the addition of CoPc. The balance between micro- and mesopores is critical for carbon-based catalyst materials to facilitate the diffusion of reactant species during the ORR/OER processes [38]. Mesopores are considered to be favourable for electrocatalysis, but increasing mesopores tends to decrease the surface area of the catalyst materials, which leads to depletion in active sites. Hence, as-prepared materials are suitable for ORR/OER due to the balanced micro/mesopore structure [39].

Next, these materials were analysed by SEM to study the surface morphology; the obtained images are shown in Figure 2. All materials possess relatively homogenous morphology with interconnected porous carbon structures. Doping with CoPc leads to mild changes in the morphology. The surface structure of the materials appears to be more compact with the increased amount of CoPc (see Figure 2 inset). SEM-EDX-assisted elemental mapping was performed to see the distribution of dopants in the catalysts and detect any potential agglomeration of Co (shown in Appendix A). As expected, no agglomeration of elements was detected in the element mapping, which corroborates the XRD results. The bulk concentration of elements was determined by SEM-EDX (Table 2). The analyses show the high nitrogen and cobalt concentration, which indicates that doping with CoPc was successful. Additionally, oxygen functionalities are from carbon–oxygen species and metal oxides, which are believed to be suitable for OER [40,41].

The XPS measurements were conducted to identify the surface elemental composition and different states of dopants that existed in the prepared catalysts (Figure 3). The surface of the catalysts exhibited the presence of C, N, O, and Co, as indicated by the XPS survey spectrum (Table 3). The nitrogen and oxygen contents were similar in all prepared catalysts, with the only variation found in the Co content, which was expected due to the different CoPc amounts used in the catalyst preparation. Different nitrogen species, including pyridinic-N, pyrrolic-N, graphitic-N, and metal-coordinated N (M-Nx), were detected on the catalyst’s surface, as revealed by the deconvoluted high-resolution N 1s XPS peak (Figure 3a,c inset). Among all three materials, pyridinic-N and pyrrolic-N exhibited the highest prominence, followed by graphitic-N and M-Nx. (Figure 3d). These results indicate that the doping with 2-methylimidazole and CoPc increased the concentration of different N-species to promote the ORR kinetics. Transition metal-N_x_ centres are also considered to be accountable for high ORR activity via 4e^−^ or 2 × 2e^−^ pathways [42]. It is noteworthy that the Co-N-PC-2 catalyst consisted of relatively higher percentages of pyridinic-N and M-Nx compared to Co-N-PC-1 (Figure 3d), which could be beneficial for the higher electrocatalytic activity. The deconvolution of the O 1s peak from the XPS spectra reveals the presence of cobalt oxide, carboxyl, hydroxyl groups, and chemisorbed water (Appendix A). These oxygen species were reported to enhance the electrocatalytic activity of the material [43,44]. The increase in cobalt content leads to a higher amount of oxygen species, including metal oxide, carboxyl, hydroxyl groups, and chemisorbed water. Metal oxides can form during the pyrolysis of MN_4_ macrocycles (Appendix A), and their presence has been shown to enhance the ORR/OER activity in alkaline media [41,45,46,47]. Further, the individual components of the Co2p XPS spectra were resolved to quantify the oxidation state of Co and determine the composition of each Co species. Appendix A shows two prominent peaks at 780.4 and 796 eV, ascribed for Co2p_3/2_ and Co2p_1/2_, respectively. The deconvoluted peaks observed at 778 and 780.4 eV can be attributed to metallic Co and Co-N in the Co2p_3/2_ spectrum, respectively. This further assures the presence of Co-Nx centres. For CoN-PC-1, the concentration of Co-N (0.11 at%) is lower than CoN-PC-2 (0.23 at%).

### 3.2. Electrochemical Properties of the Catalysts

Firstly, the ORR results of CoN-PC-2 catalysts prepared at different pyrolysis temperatures (700, 800, and 900 °C) are presented in Appendix A. The observed outcomes suggest that pyrolysis temperatures have an impact on the ORR activity. The catalyst synthesised at 800 °C demonstrated superior ORR activity, exhibiting a half-wave potential (*E*_1/2_) of 0.81 V vs. RHE. As a result, 800 °C was selected as the pyrolysis temperature for the preparation of other catalysts.

Further, the effect of CoPc was studied on the ORR and OER activity using the RDE method. As expected, PC-800 exhibited the least electrocatalytic activity for the ORR, as evidenced by an onset potential for oxygen reduction (*E*_onset_) of 0.79 V, where the ORR current density reaches −0.1 mA cm^−2^. Moreover, PC-800 follows two reduction pathways during the ORR process. A substantial improvement in the electrocatalytic activity was noticed with the addition of CoPc. The overall ORR electrocatalytic activity of CoN-PC-1 and CoN-PC-2 was almost the same in the RDE experiment, as both had *E*_onset_ and *E*_1/2_ values of 0.88 V and 0.81 V, respectively. Commercial Pt/C (*E*_1/2_ = 0.85 V) was also compared using the RDE method, and the results are shown in Figure 4a and Table 4. The OER was evaluated in terms of *E*_j=10_. CoN-PC-2, with an *E*_j=10_ value of 1.70 V, exhibited the best OER activity among the as-prepared catalysts (Figure 4b). A slightly higher *E*_j=10_ value was observed for CoN-PC-1 (1.73 V), and PC-800 could not attain comparable activity. This trend is expected in these catalysts, as PC-800 does not have any metal-containing active sites for OER and an increased OER activity was observed for catalysts with higher metal contents, which can provide more active sites for OER.

To investigate the ORR pathways, linear sweep voltammograms (LSV) were measured at different rotation rates (360–3100 rpm) using RDE in O_2_-saturated 0.1 M KOH electrolytes (Appendix A). Appendix A display the Koutecky–Levich (K-L) plots at different potentials for O_2_ reduction, while the K-L equation is employed to determine the electron transfer number (n) during the ORR process [48]:(1)$$j^{-1}=j_{k}^{-1}+j_{d}^{-1}$$
(2)$$j_{d}=0.62nFD_{0}^{2/3}v^{-1/6}C_{0}\omega^{1/2}$$

In the given equations, *j*, *j_k_*, and *j_d_* represent the experimentally measured, kinetic, and diffusion-limited current densities, respectively. *F* corresponds to the Faraday constant (96,485 C mol^−1^), *C*_0_ denotes the bulk concentration of O_2_ in a 0.1 M KOH solution (1.2 × 10^−6^ mol cm^−3^), *v* represents the kinematic viscosity of the electrolyte solution (0.01 cm^2^ s^−1^), *D*_0_ is the diffusion coefficient of O_2_ in a 0.1 M KOH solution (1.9 × 10^−6^ cm^2^ s^−1^), and *ω* stands for the electrode rotation rate (rad s^−1^). The linearity of K-L plots suggests the first-order kinetics for ORR (Appendix A), and the *n* value was determined to be close to four (~3.5) for CoN-PC (Appendix A). Table 4 provides a comparison with previously published ORR/OER electrocatalysts, and the comparison clearly demonstrates the efficiency of the catalysts employed in this study in terms of *E*_1/2_ and *E*_j=10_ values.

**Table 4 materials-16-05105-t004:** Oxygen reduction and evolution reaction results of the prepared catalysts and comparison with the literature.

Catalyst	*E*_onset_/V	*E*_1/2_/V	*E*_j=10_/V	Refs.
PC-800	0.79	-	-	This work
CoN-PC-1	0.88	0.81	1.73	
CoN-PC-2	0.88	0.81	1.70	
Pt/C	0.97	0.85	-	
RuO_2_	-	-	1.60	
Co/CNT/MCP-850	0.94	0.80	1.50 *	[49]
Co@N-CNTF-2	0.91	0.81	1.58 *	[50]
Co-ZIF_1.5_/10CNF_2_	0.93	0.85	1.62	[51]
Co/CNFs(1000)	1.01	0.896	1.55 *	[52]
Co_2_P/CoN-in-NCNTs	0.96	0.85	1.64	[53]
CoOOH-PHCS	-	0.74	1.57	[54]
CoNC@LDH	-	0.84	1.47	[55]
LSTFO/NCNT	0.99	0.86	1.62	[56]

* measured in 1 M KOH solution.

The cyanide poisoning test was also carried out to evidence the presence of the M-Nx active sites in the catalyst materials. In principle, cyanide ions strongly bind to the M-Nx sites, impairing the catalysts’ ORR activity [57]. The LSV curves were obtained in 0.1 M KOH solution containing 10 mM NaCN. It is clear from Figure 5 that the catalysts’ ORR activity significantly dropped with the addition of cyanide ions. These test findings also confirm the catalysts’ M-Nx centres, which were identified in the XPS analysis.

Despite the high electrocatalytic activity, good stability is important for electrocatalysts in metal-air batteries due to the high operational voltage. To evaluate the electrochemical stability of the synthesised catalysts, 10,000 repeated potential cycles in an O_2_-saturated 0.1 M KOH solution were performed (Figure 6). CoN-PC-1 revealed good stability with only 14 mV shift in *E*_1/2_, while no difference in *E*_1/2_ was observed for CoN-PC-2. This is attributed to the hierarchical porous structures in the CoN-PC catalysts, which mitigate the metal dissolution and aggregation during the vigorous electrocatalytic process by stabilising metal-Nx active sites and accelerating the electron transfer process [58]. Therefore, the hierarchical porous structure and optimised metal concentration collectively promoted excellent electrochemical stability. For the purpose of evaluating the stability of the OER, chronoamperometric measurements were conducted at a potential of 1.7 V in an Ar-saturated 0.1 M KOH solution (Appendix A). After 3 h, CoN-PC-2 and CoN-PC-1 maintained 79% and 51% of the initial OER activity, respectively. Higher stability of CoN-PC-2 was ascribed to the higher content of Co, which can help to stabilise the carbon matrix and mitigate the loss of active sites.

### 3.3. RZAB Test Results

The performance of the developed CoN-PC-1 and CoN-PC-2 catalyst materials in terms of oxygen bifunctional electrocatalysis at the cathode was evaluated by assembling and assessing rechargeable zinc-air batteries. The discharge polarisation curves and power density curves for catalysts are shown in Figure 6a. Remarkably, CoN-PC-1 and CoN-PC-2 catalysts showed open circuit voltage (OCV) values as high as 1.45 and 1.47 V, respectively. As shown in Figure 7a, CoN-PC-2 had a greater maximum power density of 158 mW cm^−2^, slightly higher than that of CoN-PC-1 (155 mW cm^−2^). Furthermore, a galvanostatic charge-discharge cycling test (10 min/cycle) carried out at a current density of 5 mA cm^−2^ was used to assess the durability of both catalysts (Figure 7b). Under the same circumstances, the CoN-PC-2 catalyst exhibited an excellent cycling performance in comparison with CoN-PC-1. Interestingly, both CoN-PC-1 and CoN-PC-2 catalysts had an initial voltage gap of around 0.9 V between their charge and discharge potentials. However, after 45 h of cycling, the voltage gap of CoN-PC-2 increased to 1.08 V, while CoN-PC-1-based RZAB failed two times within 30 h. The high stability of CoN-PC-2 is attributed to a higher concentration of metal, which can provide sustainable active sites for ORR/OER and lower carbon decomposition in highly alkaline media. CoN-PC-2 exhibits comparable RZAB performance to the reported RZABs (Table 5). Nonetheless, comparing different RZABs is not always straightforward due to many factors, including electrode distance, air flow, Zn anode, the volume of the electrolyte, etc. [11].

## 4. Conclusions

In summary, cobalt- and nitrogen-doped porous carbon was prepared using phloroglucinol-formaldehyde networks and 2-methylimidazole and cobalt phthalocyanine as precursors. The prepared catalyst CoN-PC-2 demonstrated good bifunctional electrocatalytic activity towards the ORR and OER in alkaline media with *E*_1/2_ = 0.81 V and *E*_j=10_ = 1.70 V. The controlled synthesis of phloroglucinol-formaldehyde networks resulted in the mixture of micro- and mesoporous carbon structures, which are considered to play a vital role in enhancing the overall electrocatalytic activity. Furthermore, the higher cobalt content and hierarchical porous structure of CoN-PC-2 contribute towards the excellent RZAB performance in terms of OCV (1.47 V), *P*_max_ (158 mW cm^−2^), and the excellent stability of 45 h. Overall, the present work provides new insights for the preparation of active electrocatalysts for rechargeable zinc-air batteries through a facile and simple synthesis route.

## Figures and Tables

**Figure 1 materials-16-05105-f001:**
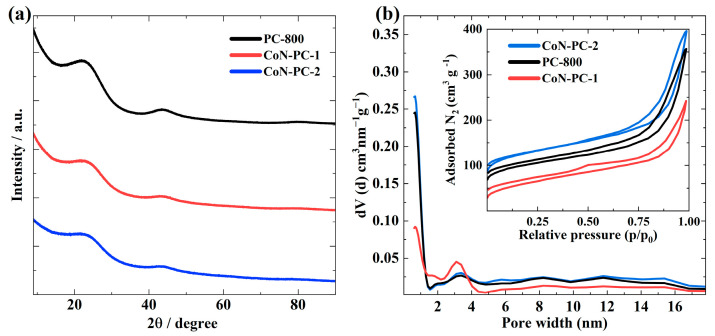
(**a**) XRD patterns and (**b**) pore size distribution and nitrogen adsorption-desorption isotherms (inset) of the prepared catalysts.

**Figure 2 materials-16-05105-f002:**
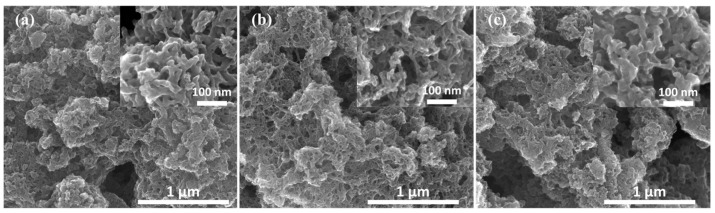
SEM images (inset: higher magnification images) of (**a**) PC-800, (**b**) CoN-PC-1, and (**c**) CoN-PC-2.

**Figure 3 materials-16-05105-f003:**
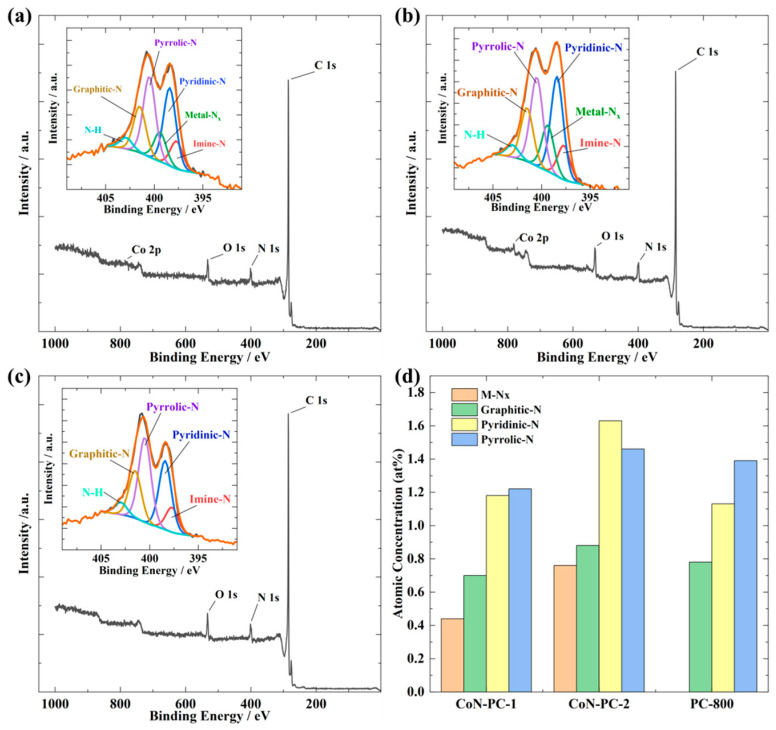
XPS survey spectra and N 1s high-resolution spectra (inset) of (**a**) CoN-PC-1, (**b**) CoN-PC-2, and (**c**) PC-800 and (**d**) graphical representation of nitrogen species surface concentration in prepared electrocatalysts.

**Figure 4 materials-16-05105-f004:**
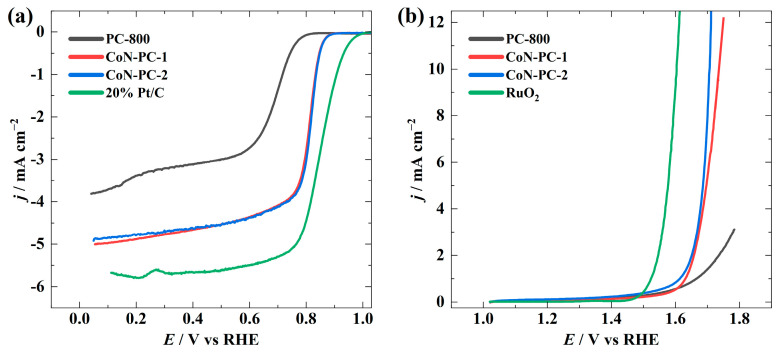
(**a**) ORR and (**b**) OER polarisation curves in 0.1 M KOH solution at 1900 rpm and 10 mV s^−1^.

**Figure 5 materials-16-05105-f005:**
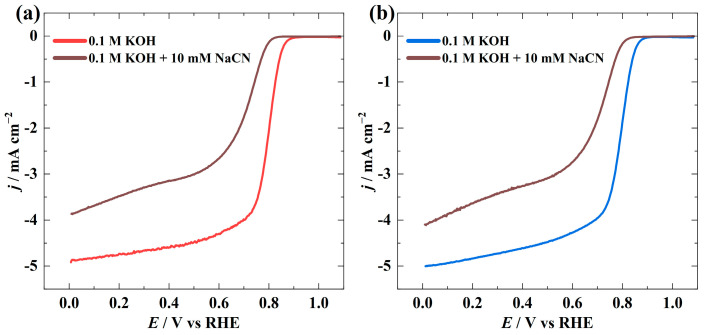
ORR polarisation curves for (**a**) CoN-PC-1 and (**b**) CoN-PC-2 catalysts in O_2_-saturated 0.1 M KOH solution supplemented with 10 mM NaCN at 1900 rpm (*v* = 10 mV s^−1^).

**Figure 6 materials-16-05105-f006:**
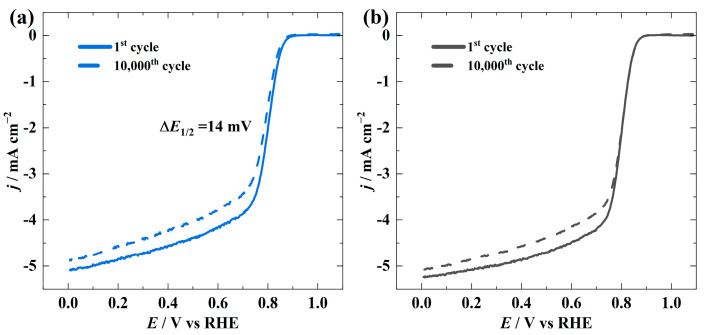
Polarization curves of ORR before and after stability test for (**a**) CoN-PC-1 and (**b**) CoN-PC-2 catalysts in O_2_-saturated 0.1 M KOH solution at 1900 rpm with a scan rate of 10 mV s^−1^.

**Figure 7 materials-16-05105-f007:**
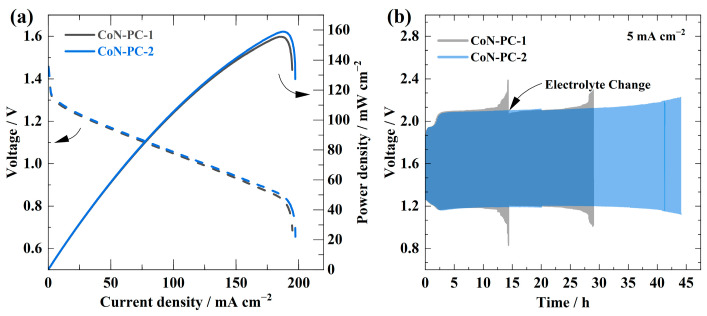
(**a**) Polarisation and power density curves of CoN-PC-1 and CoN-PC-2 catalysts for discharging. (**b**) Stability test for RZABs using CoN-PC-1 and CoN-PC-2 catalysts by galvanostatic discharge-change cycling at 5 mA cm^−2^.

**Table 1 materials-16-05105-t001:** Nitrogen physisorption results of the prepared catalysts.

Catalyst	*S*_BET_ (m^2^ g^−1^)	*V*_tot_ (cm^3^ g^−1^)	*V*_µ_ (cm^3^ g^−1^)
PC-800	449	0.54	0.15
CoN-PC-1	464	0.43	0.16
CoN-PC-2	325	0.35	0.12

**Table 2 materials-16-05105-t002:** Catalysts’ elemental composition determined via SEM-EDX (wt.%).

Catalyst	C	N	O	Co
PC-800	84.4	10.3	5.3	-
CoN-PC-1	82.0	9.1	7.7	1.2
CoN-PC-2	79.9	10.3	7.7	2.2

**Table 3 materials-16-05105-t003:** Elemental surface composition of electrocatalysts determined by XPS (at%).

Catalyst	C	N	O	Co
PC-800	91.3	4.2	4.6	-
CoN-PC-1	91.2	4.2	4.3	0.4
CoN-PC-2	88.0	5.5	5.7	0.8

**Table 5 materials-16-05105-t005:** Comparison of transition metal-based catalysts for RZABs.

Catalyst	Loading/mg cm^−2^	OCV/V	*P*_max_/mW cm^−2^	Refs.
CoN-PC-1	2.0	1.45	155	This
CoN-PC-2	2.0	1.47	158	work
FeNi-COP-800	2.0	-	64	[59]
Fe-Sas/NPS-HC	1.0	1.45	195	[60]
CoO-NSC-900	1.0	1.40	~68	[61]
FeNi/N-GPCM	-	1.473	321	[62]
V_0_-CMON@NCN	-	1.47	143.7	[63]
CoNC@LDH	1.5	-	173	[55]
LSTFO/NCNT	1.0	1.54	109.6	[56]

## Data Availability

The data presented in this study are available on request from the corresponding author (kaido.tammeveski@ut.ee).

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
