# Peer review of "Cobalt Phthalocyanine-Doped Polymer-Based Electrocatalyst for Rechargeable Zinc-Air Batteries"

_materials, 2023, doi:10.3390/ma16145105_

Round 1

Reviewer 1 Report

In this work, the authors prepared the Co/N-doped porous carbon derived from phloroglucinol-formaldehyde polymer networks with 2-methyl imidazole and cobalt phthalocyanine precursors, which exhibits good activity and stability for Zn-air battery. The underlying performance origins were also revealed. Overall, this work can be accepted after well addressing my comments below.

1.      Besides the XRD, Raman results of the samples should be performed to identify the C/N-based materials, which will help compare the degree of graphitization in samples.

2.      Oxygen species are extremely important for OER and ORR reactions. The authors should conduct the O 1s XPS analysis for their samples. Also, the fitting of O 1s XPS is necessary. Please refer to these papers about O1s XPS fitting to demonstrate the performance origins for OER (DOI: 10.1039/C9TA06020K) and ORR (DOI: 10.1002/chem.201705675).

3.      Also, the analysis and comparison of Co 2p XPS for the samples are necessary and helpful to reveal the performance origins.

4.      The OER stability should be provided, which is missing in this version.

 Minor editing of English language required

Reviewer 2 Report

In this work, the authors prepared Co and N-doped porous carbon materials derived from poly-  condensation of phloroglucinol with formaldehyde via the formation of a polymer frame- 73 work and cobalt phthalocyanine and 2-methylimidazole. The material showed interesting electrochemical performance for oxygen electrocatalysis and Zn-air battery. Detailed chemical and electrochemical performances are characterized. The work is interesting and the topic is suitable for materials. Before accepting for publication, the reviewer suggests to revise the following points:

1.       In the introduction, the authors mentioned that the , porous carbon materials synthesised using these methods suffer severe degradation at high voltage conditions in RZABs due to their insufficient graphitisation levels. Therefore, the stability of the prepared electrode at high electrode potential like 0.9 V vs. RHE under ORR and 1.7 V under OER condition should be performed. The stability of the sample that was treated at 700 oC and 900 oC may be also tested for comparison. By the way, the reason of the reduced performance of catalyst after stability study should be also examined and discussed.

2.       Its seems that the Co elenment in the catalyst also played important role. No any chemical analysis on Co?

3.       The performance is not compared with the literature value, references with similar material and electrochemical system like Adv Mater 33 (2021) e2008606 ; Mater Today Nano 21 (2023) 100287; Nano Research 16 (2023) 80 could be added to revised manuscript.
